# Different levels of hypoglycemia in patients with type 2 diabetes, their achieved mean HbA1c vs. all-cause and cardiovascular mortality

**Seng-Wei Ooi[1], Shu-Tin Yeh[1], Ya-Hui Chang[2,3], Chung-Yi Li[2,4,5], Hua-Fen Chen** [1,6] *

**1** Department of Endocrinology, Far Eastern Memorial Hospital, New Taipei City, Taiwan, **2** Department of Public Health, College of Medicine, National Cheng Kung University, Tainan, Taiwan, **3** Department of Surgery, Massachusetts General Hospital, Boston, MA, United States of America, **4** Department of Public Health, College of Public Health, China Medical University, Taichung, Taiwan, **5** Department of Healthcare Administration, College of Medical and Health Science, Asia University, Taichung, Taiwan, **6** School of Medicine and Department of Public Health, College of Medicine, Fujen Catholic University, New Taipei City, Taiwan

* hfchen@mail.femh.org.tw

**Data Availability Statement:** The data sets analyzed during the current study are not publicly available because the restrictions are imposed by the Research Ethics Review Committee of Far

## Abstract

### Aim

In patients with type 2 diabetes (T2D), levels of hypoglycemia and their risk of mortality are not well understood. The aim of this study was to ascertain the correlation among disparate levels of hypoglycemia and patients with T2D's achieved mean glycated hemoglobin A1c (HbA1c) with all-cause and cardiovascular mortality.

### Methods

27,932 T2D patients taking hypoglycemic medications at outpatient visits for more than 6 months between 2008 and 2018 were linked to Taiwan's National Death Registry. We determined the respective mortality rates with Poisson assumption, and explored the relative risks of all-cause and cardiovascular mortality according to dissimilar levels of hypoglycemia with their achieved mean HbA1c by Cox proportional hazard regression model with adjustment of potential confounders.

### Results

T2D patients with level 3 hypoglycemia had the highest rates of all-cause and cardiovascular mortality. Compared with those who never encountered hypoglycemia, study subjects with level 1 and level 2 hypoglycemia did not show excessive risks of either all-cause or cardiovascular mortality. Only those with level 3 hypoglycemia revealed marginal risk of all-cause (Hazard ratio [HR]: 1.18; 95% Confidence Interval [CI] 1.04–1.33) but not cardiovascular mortality (HR: 1.16; 95% CI 0.88–1.53). In T2D patients with hypoglycemia, only those with mean HbA1c ≥9.0% increased all-cause mortality in level 3 hypoglycemia, and cardiovascular mortality in level 1 hypoglycemia.

Eastern Memorial Hospital. Although we cannot share the data publicly, the minimal datasets are available upon request from the Research Ethics Review Committee of Far Eastern Memorial Hospital repository (via: https://www.femh-irb.org/index.php/procedure). Application to the Research Ethics Review Committee of Far Eastern Memorial Hospital (contact via irb@mail.femh.org.tw, reference number 108008-F) is available for researchers.

**Funding:** This study received financial support from Far Eastern Memorial Hospital in the form of grants (FEMH-2021-C-034, FEMH-2022-C-015, and FEMH-2023-C-082). No additional external funding was received for this study. The funder had no role in study design, data collection and analysis, decision to publish, or preparation of the manuscript.

**Competing interests:** The authors have declared that no competing interests exist.

## Conclusions

Elevated risk of all-cause mortality was exclusively found in patients with level 3 hypoglycemia. In T2D patients with hypoglycemia, mean HbA1c $\geq$ 9% increased all-cause or cardiovascular mortality. Aggressive treatment of accompanying serious illness in severe hypoglycemia may help reduce mortality in patients with T2DM.

## Introduction

Large prospective randomized controlled trials [1] showed intensive glycemic control was effective to prevent complications in patients with type 2 diabetes (T2D), but hypoglycemia is the major limiting factor in the optimal glycemic management of diabetes. Although some of the previous studies reported that hypoglycemia might be associated with increased all-cause [2–6] and cardiovascular mortality [6], other studies refuted such associations [7, 8]. Most of the hypoglycemia events were collected by self-report [2, 7, 8] or diagnostic codes [3–6] rather than documented laboratory values so that the classification of hypoglycemia could not be as specific as that defined by the International Hypoglycemia Study Group [9]. Lack of information about medication use, status of blood glucose and lipid control, which might also have confounded the all-cause and cardiovascular mortality, is also the major limitations of some of the above studies [2, 3].

In both the Action to Control Cardiovascular Risk in Diabetes (ACCORD) trial [10] and the Action in Diabetes and Vascular Disease: PreterAx and Diamicron MR Controlled Evaluation (ADVANCE) trial [11], symptomatic, severe hypoglycemic events were commoner in the intensive glycemic control group. However, in a retrospective epidemiological analysis of data from the ACCORD trial, annual mortality among patients who reported severe hypoglycemia was higher in the group receiving standard treatment rather than in the group receiving intensive treatment [10]. Neither a temporal nor dose–response relationship between hypoglycemia and mortality observed in the ACCORD trial produces the uncertainty about the causal relationship between hypoglycemia and all-cause or cardiovascular mortality. In our previous study [12], higher risks associated with low mean glycated hemoglobin A1c (HbA1c) and all-cause mortality were attenuated by adjustment of medications and comorbidities, but the relationship between mean HbA1c and hypoglycemia patients has not been clearly elucidated.

This study aimed to determine the association between different levels of hypoglycemia classified by the International Hypoglycemia Study Group [9] with all-cause and cardiovascular mortality in T2D patients at Far Eastern Memorial Hospital (FEMH). We also did statistical analyses with further stratification of the calculated mean HbA1c levels in T2D patients with hypoglycemia to see whether the achieved mean HbA1c levels would affect the risk of mortality in each level of hypoglycemia.

## Materials and methods

### Study subjects

FEMH, a tertiary medical center, has set up its electronic medical database of out-patient and emergency visits since 2001 and 2008, respectively. The details of our out-patient electronic database have been described in previous studies [12–14]. In this study, we extended our database to emergency visits electronic records. This study was approved by the Research Ethics

Review Committee, Far Eastern Memorial Hospital (108008-F) without the necessity of informed consent.

44,910 patients with diagnosis of diabetes (ICD-9-CM: 250.xx or ICD-10-CM: E10 or E11) from out-patient database (2008–2018) were identified. We excluded 3,251 patients not taking any hypoglycemic medication, 13,467 patients with duration of total out-patient visits at FEMH not more than 6 months, 36 patients with no PIN, and 224 patients with a diagnosis of type 1 diabetes. The number of study subjects in this study altogether were 27,932 patients.

### Follow-up, covariates and study endpoints

From the out-patient medical records, the information of oral hypoglycemic agents, insulin, antihypertensive and antilipid medications were collected during the study period as stated in our previous studies [12–14]. We also explored each study participant's comorbidities and complications as described in S1 Table [15]. The laboratory values analyzed in this study were HbA1c, fasting plasma glucose (FPG), low density lipoprotein (LDL), high density lipoprotein (HDL), triglycerides (TG), creatinine (Cr), and hemoglobin (Hb) levels. eGFR was calculated according to the equation previously reported [16]. We chose the highest value of HbA1c, FPG, LDL, HDL, TG, Cr, and the lowest value of Hb for each quarter, and computed mean of the annual mean for the whole study period. Anemia was defined as mean Hb <13 and <12 g/dL for men and women [17], respectively.

If the fasting or postprandial plasma or point-of-care (POC) glucose values of the patients with T2DM detected at out-patient clinic between 2008 and 2018 were ≥54-<70 mg/dL and <54 mg/dL [9], they were defined as level 1 and level 2 hypoglycemia, respectively. The dates of level 1 and 2 hypoglycemia were the first date of incidents occurring at out-patient visit during the study period. We determined those patients who came to the emergency department with the chief complaint of hypoglycemia in association with either glucose administration or blood glucose level ≤70mg/dL documented in the electronic medical record as level 3 hypoglycemia. The date of level 3 hypoglycemia was the first date of an emergency visit with a related hypoglycemia event. If the patient had hypoglycemia for more than one occasion or one level of hypoglycemia during the study period, the index date was set to the first day of the hospital visit due to the most severe level of hypoglycemia. Control group was all other patients with T2D who had never encountered any level of hypoglycemia during the study period. We set the first date of oral or parenteral antidiabetic agent prescription in FEMH as the index date of our study subjects. If it was before Jan 1, 2008, the index date was Jan 1, 2008. The difference between the index date and the date of birth was determined as the age of each T2D patient.

By the linkage of FEMH's electronic database to Taiwan's National Death Registry [18], we evaluated all-cause and cardiovascular mortality (ICD-9-CM: 390–392, 393–398, 410–414, 420–429, 430–438 or ICD-10-CM: I01-I02.0, I05-I09, I20-25, I27, I30-I52, I60-I69) of our study subjects. If the patient did not encounter death during the study period and his/ her laboratory value was detected beyond Dec 31, 2018, they were censored at the end of the study (i.e. Dec 31, 2018). For other patients, the dates of the last out-patient visit in FEMH were set to be their censored dates.

### Statistical analysis

We used Poisson assumption to calculate all-cause and cardiovascular mortality rates of patients with T2DM by different levels of hypoglycemia. In addition, we determined the relationship of patients' various levels of hypoglycemia with the relative risks of all-cause or cardiovascular mortality by applying Cox proportional hazard regression model with adjustment of age and sex in model 1; age, sex, medications use, comorbidities and laboratory results in

model 2. We also determined the risks of all-cause and cardiovascular mortality by different categories of HbA1c (<6.0, 6.0–6.9, 7.0–7.9, 8.0–8.9, ≥9.0%) in various levels of hypoglycemia of our T2D patients in fully adjusted models. Our statistical analyses were performed with SAS (version 9.4; SAS Institute, Cary, NC).

## Results

### Patients' baseline characteristics

Table 1 shows the patients' baseline characteristics according to the levels of hypoglycemia. T2D patients aged 50–69 years comprised of higher proportions of level 1 and level 2 hypoglycemia, but in level 3 hypoglycemia, half of the patients were those aged >69 years. More men suffered from level 1 and level 2 hypoglycemia while more women were sent to the emergency department for level 3 hypoglycemia. SU and DPP4i prescriptions were the highest in patients with level 1 hypoglycemia. On the other hand, in those patients with level 3 hypoglycemia, metglitinides and insulin use were the greatest while SGLT2i was the least prescribed antidiabetic agent. The percentage of comorbidities and complications were the highest in T2D patients with level 3 hypoglycemia.

Patients with hypoglycemia had higher mean HbA1c and FPG, but lower mean eGFR, Hb, and LDL. The proportion of patients with mean HbA1c <6.0% were similar among patients with various levels of hypoglycemia or no hypoglycemia. The greatest percentage of patients with all levels of hypoglycemia had their mean HbA1c controlled around 7.0–7.9%, and a higher portion of patients with mean HbA1c ≥9.0% were more likely to develop level 2 and 3 hypoglycemia.

After a follow-up period about 5 years, T2D patients with hypoglycemia were more likely to encounter higher all-cause and cardiovascular mortality. Patients with level 3 hypoglycemia had the most elevated proportion of all-cause (51.62%) and cardiovascular (9.35%) mortality compared to those without hypoglycemia (14.92% and 2.95%, respectively).

### Overall rates and relative risks of all-cause and cardiovascular mortality by different levels of hypoglycemia in patients with type 2 diabetes

The lowest overall all-cause mortality was observed in T2D patients with no hypoglycemia (33.27/1,000 patient-year [PY] and level 1 hypoglycemia (35.04/1,000 PY). Those with level 2 hypoglycemia had higher all-cause mortality rates (54.87/1,000 PY), and T2D patients with level 3 hypoglycemia had the most elevated all-cause mortality rates (96.59/1,000 PY).

Compared with patients without hypoglycemia, those who had level 1 hypoglycemia did not increase the risk of all-cause mortality after adjustment of covariates in model 1 (Hazard ratio [HR]: 0.94; 95% confidence interval [CI]: 0.86–1.04) and model 2 (HR: 0.91; 95% CI: 0.81–1.02). In patients with level 2 hypoglycemia, the higher risk of all-cause mortality detected in model 1 (HR: 1.30; 95% CI 1.14–1.48) became statistically inconsequential with additional adjustment of medications, comorbidities, complications, and laboratory results in model 2 (HR: 1.00; 95% CI 0.86–1.16). The highest risks noted in those who had history of level 3 hypoglycemia (HR: 1.63; 95% CI 1.47–1.80) persisted in model 2, but the HR became much attenuated (HR: 1.18; 95% CI 1.04–1.33) (Table 2).

Cardiovascular mortality rates were comparable between patients with no hypoglycemia (6.58/1,000 PY) and level 1 hypoglycemia (6.55/1,000 PY), and the rates became higher in patients who encountered level 2 and level 3 hypoglycemia (9.83 and 17.50/1,000 PY, respectively). Patients with level 1 and level 2 hypoglycemia did not reveal any elevated risks of cardiovascular mortality in both model 1 and model 2. Similarly, the higher risks of

**Table 1. Characteristics of type 2 diabetic patients according to different levels of hypoglycemia.**

| Variables[1,2] | No hypoglycemia | | Hypoglycemia Level 1 | | Hypoglycemia Level 2 | | Hypoglycemia level 3 | |
|---|---|---|---|---|---|---|---|---|
| | n | % | n | % | n | % | n | % |
| **General Characteristics** | | | | | | | | |
| Age | | | | | | | | |
| <50 | 6,065 | 25.08 | 486 | 22.25 | 130 | 18.73 | 65 | 7.51 |
| 50–69 | 13,464 | 55.67 | 1,248 | 57.14 | 374 | 53.89 | 362 | 41.80 |
| >69 | 4,658 | 19.26 | 450 | 20.60 | 190 | 27.38 | 439 | 50.69 |
| Mean age (±SD) | 58.40 | 13.30 | 59.20 | 13.05 | 6 1.45 | 13.35 | 68.38 | 11.95 |
| Sex | | | | | | | | |
| Women | 10,642 | 44.00 | 1,015 | 46.47 | 332 | 47.84 | 469 | 54.16 |
| Men | 13,546 | 56.00 | 1,169 | 53.53 | 362 | 52.16 | 397 | 45.84 |
| **Medications** | | | | | | | | |
| Oral Antidiabetic agents | 23,550 | 97.36 | 2,054 | 94.05 | 612 | 88.18 | 801 | 92.49 |
| Sulphonylureas | 13,956 | 57.70 | 1,676 | 76.74 | 482 | 69.45 | 589 | 68.01 |
| Meglitinides | 2,843 | 11.75 | 502 | 22.99 | 210 | 30.26 | 383 | 44.23 |
| Metformin | 21,206 | 87.67 | 1,768 | 80.95 | 466 | 67.15 | 547 | 63.16 |
| Thiazolidinediones | 2,463 | 10.18 | 367 | 16.80 | 112 | 16.14 | 121 | 13.97 |
| Alpha-glucosidase inhibitors | 2,855 | 11.80 | 380 | 17.40 | 113 | 16.28 | 158 | 18.24 |
| Dipeptidyl peptidase-4 inhibitors | 12,722 | 52.60 | 1,321 | 60.49 | 386 | 55.62 | 486 | 56.12 |
| Sodium-glucose cotransporter-2 inhibitors | 2,940 | 12.15 | 236 | 10.81 | 48 | 6.92 | 16 | 1.85 |
| Insulin | 4,814 | 19.90 | 999 | 45.74 | 471 | 67.87 | 441 | 50.92 |
| Antihypertensives | 19,108 | 79.00 | 1,876 | 85.90 | 630 | 90.78 | 786 | 90.76 |
| Antilipids | 18,003 | 74.43 | 1,721 | 78.80 | 536 | 77.23 | 569 | 65.70 |
| **Comorbidities and complications** | | | | | | | | |
| Coronary artery disease | 6,460 | 26.71 | 624 | 28.57 | 198 | 28.53 | 288 | 33.26 |
| Heart failure | 2,210 | 9.14 | 263 | 12.04 | 112 | 16.14 | 204 | 23.56 |
| Hypertensive disease | 16,750 | 69.25 | 1,648 | 75.46 | 574 | 82.71 | 710 | 81.99 |
| Cerebrovascular disease | 3,191 | 13.19 | 381 | 17.45 | 142 | 20.46 | 216 | 24.94 |
| Peripheral artery disease | 447 | 1.85 | 60 | 2.75 | 28 | 4.03 | 46 | 5.31 |
| Diabetic nephropathy | 5,423 | 22.42 | 969 | 44.37 | 385 | 55.48 | 476 | 54.97 |
| Diabetic retinopathy | 6,314 | 26.10 | 1,002 | 45.88 | 354 | 51.01 | 398 | 45.96 |
| Diabetic neuropathy | 2,919 | 12.07 | 448 | 20.51 | 171 | 24.64 | 225 | 25.98 |
| **Laboratory results** | | | | | | | | |
| **Mean** | | | | | | | | |
| HbA1c (%) (SD) | 7.61 | 1.32 | 7.80 | 1.22 | 7.96 | 1.22 | 7.89 | 1.43 |
| <6 | 859 | 3.59 | 75 | 3.43 | 24 | 3.46 | 32 | 3.74 |
| 6.0–6.9 | 8,164 | 34.16 | 486 | 22.25 | 124 | 17.87 | 210 | 24.56 |
| 7.0–7.9 | 7,876 | 32.96 | 816 | 37.36 | 238 | 34.29 | 272 | 31.81 |
| 8.0–8.9 | 3,803 | 15.91 | 486 | 22.25 | 189 | 27.23 | 183 | 21.40 |
| ≥9 | 3195 | 13.37 | 321 | 14.70 | 119 | 17.15 | 158 | 18.48 |
| FPG (mg/dL) (SD) | 144.62 | 38.52 | 138.81 | 37.38 | 144.25 | 42.67 | 151.52 | 47.26 |
| LDL (mg/dL) (SD) | 105.71 | 24.86 | 101.85 | 22.64 | 99.01 | 24.11 | 98.71 | 25.88 |
| HDL (mg/dL) (SD) | 45.82 | 12.31 | 45.70 | 12.69 | 45.97 | 14.21 | 45.37 | 13.93 |
| TG (mg/dL) (SD) | 172.46 | 132.55 | 163.72 | 98.90 | 157.15 | 100.41 | 153.72 | 93.50 |
| eGFR (ml/min/m$^2$) (SD) | 57.20 | 8.11 | 52.44 | 13.07 | 48.02 | 16.58 | 44.87 | 16.46 |
| ≥60 | 15,507 | 71.42 | 1,063 | 48.85 | 238 | 34.49 | 183 | 23.37 |
| <60 | 6,205 | 28.58 | 1,113 | 51.15 | 452 | 65.51 | 600 | 76.63 |

*(Continued)*

**Table 1.** (Continued)

| Variables[1,2] | No hypoglycemia | | Hypoglycemia Level 1 | | Hypoglycemia Level 2 | | Hypoglycemia level 3 | |
|---|---|---|---|---|---|---|---|---|
| | n | % | n | % | n | % | n | % |
| Hb (g/dL) (SD) | 13.44 | 2.00 | 12.58 | 2.17 | 11.91 | 2.09 | 11.33 | 1.89 |
| No anemia | 13,921 | 73.10 | 1,084 | 54.23 | 266 | 40.74 | 217 | 27.40 |
| Anemia | 5,124 | 26.90 | 915 | 45.77 | 387 | 59.26 | 575 | 72.60 |
| **Outcome** | | | | | | | | |
| All-cause Mortality | 3,610 | 14.92 | 471 | 21.57 | 240 | 34.58 | 447 | 51.62 |
| Cardiovascular-specific mortality | 714 | 2.95 | 88 | 4.03 | 43 | 6.20 | 81 | 9.35 |
| Follow-up period (year) | 4.49 | 2.89 | 6.16 | 2.92 | 6.30 | 2.78 | 5.34 | 2.84 |
| Total | 24,188 | 100.00 | 2,184 | 100.00 | 694 | 100.00 | 866 | 100.00 |

[1] Inconsistency between the total population and population summed for the individual variables was due to missing information

[2] ARV: average real variability; CV: coefficient of variation; eGFR: estimated glomerular filtration rate; FPG: fasting plasma glucose; Hb: hemoglobin; HbA1c: glycated hemoglobin; HDL: high density lipoprotein; LDL: low density lipoprotein; SD: standard deviation; TG: triglycerides

cardiovascular mortality found in those patients with level 3 hypoglycemia in model 1 (HR: 1.41; 95% CI 1.12–1.79) became statistically insignificant after adjustment of covariates in model 2 (HR: 1.16; 95% CI 0.88–1.53).

## Different levels of mean HbA1c in hypoglycemic patients with T2D and relative risks of all-cause or cardiovascular mortality

Table 3 presents the relative risks of all-cause and cardiovascular mortality rates with various levels of mean HbA1c attained during the study period in hypoglycemic T2D patients. We did not observe that low levels of mean HbA1c increase the risk of either all-cause or

**Table 2. Overall rates and relative risks of all-cause and cardiovascular mortality by different levels of hypoglycemia in type 2 diabetic patients.**

| Level of Hypoglycemia[1] | Mortality | | | Model 1 Adjusted HR (95% CI)[3] | Model 2 Adjusted HR (95% CI)[3] |
|---|---|---|---|---|---|
| | No. of patients | No. of mortality | Rates (per 1,000 patient-years) (95% CI)[2] | | |
| **All-cause mortality** | | | | | |
| No hypoglycemia | 24,188 | 3,610 | 33.27 (32.19–34.36) | 1.0 (Reference) | 1.0 (Reference) |
| Level 1 hypoglycemia | 2,184 | 471 | 35.04 (31.87–38.20) | 0.94 (0.86–1.04)[4] | 0.91 (0.81–1.02)[5] |
| Level 2 hypoglycemia | 694 | 240 | 54.87 (47.93–61.82) | 1.30 (1.14–1.48)[4] | 1.00 (0.86–1.16)[5] |
| Level 3 hypoglycemia | 866 | 447 | 96.59 (87.64–105.54) | 1.63 (1.47–1.80)[4] | 1.18 (1.04–1.33)[5] |
| **Cardiovascular mortality** | | | | | |
| No hypoglycemia | 24,188 | 714 | 6.58 (6.10–7.06) | 1.0 (Reference) | 1.0 (Reference) |
| Level 1 hypoglycemia | 2,184 | 88 | 6.55 (5.18–7.91) | 0.88 (0.71–1.09)[4] | 0.85 (0.66–1.10)[5] |
| Level 2 hypoglycemia | 694 | 43 | 9.83 (6.89–12.77) | 1.15 (0.85–1.57)[4] | 0.95 (0.67–1.35)[5] |
| Level 3 hypoglycemia | 866 | 81 | 17.50 (13.69–21.32) | 1.41 (1.12–1.79)[4] | 1.16 (0.88–1.53)[5] |

[1] Inconsistency between the total population and the population summed for the individual variables was due to missing information

[2] Based on Poisson assumption, CI = confidence interval

[3] HR = hazard ratio; CI = confidence interval

[4] Based on Cox proportional hazard regression with adjustment for general characteristics (i.e., age, and sex)

[5] Based on Cox proportional hazard regression with adjustment for the covariates in Model 1 plus medications (antidiabetic, antihypertensive, and antilipid), comorbidities, complications, and laboratory results.

**Table 3. Different levels of hypoglycemia-related relative risks of all-cause and cardiovascular mortality by various mean HbA1c in patients with type 2 diabetes.**

| HbA1c (%)[1] | Level 1 hypoglycemia Adjusted HR (95% CI)[2,3] | Level 2 hypoglycemia Adjusted HR (95% CI)[2,3] | Level 3 hypoglycemia Adjusted HR (95% CI)[2,3] |
|---|---|---|---|
| **All-cause mortality** | | | |
| <6 | 0.62 (0.37–1.03) | 0.80 (0.35–1.83) | 1.10 (0.51–2.37) |
| 6.0–6.9 | 1.0 (Reference) | 1.0 (Reference) | 1.0 (Reference) |
| 7.0–7.9 | 0.86 (0.65–1.14) | 1.41 (0.92–2.15) | 1.11 (0.81–1.52) |
| 8.0–8.9 | 1.09 (0.77–1.54) | 0.85 (0.50–1.43) | 1.22 (0.82–1.82) |
| ≥9 | 1.41 (0.94–2.11) | 1.54 (0.82–2.90) | 2.13 (1.36–3.34) |
| **Cardiovascular mortality** | | | |
| <6 | 0.41 (0.09–1.82) | NA | 2.12 (0.44–10.18) |
| 6.0–6.9 | 1.0 (Reference) | 1.0 (Reference) | 1.0 (Reference) |
| 7.0–7.9 | 1.12 (0.57–2.18) | 1.25 (0.47–3.35) | 1.04 (0.49–2.19) |
| 8.0–8.9 | 1.67 (0.75–3.73) | 0.94 (0.27–3.26) | 1.01 (0.41–2.51) |
| ≥9 | 2.92 (1.13–7.54) | 2.30 (0.52–10.11) | 2.60 (0.91–7.39) |

[1] HbA1c = glycated hemoglobin

[2] HR = hazard ratio; CI = confidence interval

[3] Based on Cox proportional hazard regression with adjustment for general characteristics (i.e., age, and sex), medications (antidiabetic, antihypertensive, and antilipid), comorbidities, complications, and laboratory results

cardiovascular mortality. Actually, any mean HbA1c achieved did not reveal a significant difference in the risk of all-cause mortality in level 1 and level 2 hypoglycemia. However, in patients with level 3 hypoglycemia, those with mean HbA1c ≥9.0% had higher risks of all-cause mortality (HR: 2.13; 95% CI 1.36–3.34). However, only those patients with mean HbA1c ≥9.0% had a significantly increased risk of cardiovascular mortality in level 1 hypoglycemia, but in all other level of hypoglycemia, the HR became inconsequential after adjustment of confounders.

## Discussions

During a mean follow-up of 5 years, T2D patients with a history of level 1 and level 2 hypoglycemia in our study did not increase the risk of all-cause and cardiovascular mortality after adjustment for confounding factors. Those with level 3 hypoglycemia, i.e., severe hypoglycemia, were associated with a higher risk of all-cause mortality even after adjustment for medications, comorbidities, complications, and laboratory results. We observed an elevated risk of all-cause mortality in patients with level 3 hypoglycemia if their mean HbA1c ≥9.0% attained during the study period. Similarly, those with mean HbA1c ≥9.0% had a higher risk of cardiovascular mortality in level 1 hypoglycemia.

There were a few studies that estimated the mortality rates of patients with hypoglycemia. In the Joint Asia Diabetes Evaluation Program of Hong Kong [7], the incidence of all-cause mortality in T2D patients with recurrent mild hypoglycemia was 10/1,000 PY, which was lower than the estimates of our T2D patients with level 1 and level 2 hypoglycemia. The authors recruited patients from private or public specialists and community clinics so that the prevalence of diabetes complications might have been lower than those of our patients from a tertiary medical center. In the analysis of the UK Clinical Practice Research Datalink database [6], all-cause and cardiovascular mortality rates for T2D patients who had a prior history of hospital admission for hypoglycemia were 132 and 38/1,000 PY, which were higher than those of our T2D patients with level 3 hypoglycemia. Patients needed to be hospitalized might have

had more severe underlying illness, and subsequently might have had higher mortality rates in the UK study compared to ours. In a study based on Taiwan's National Health Research Database [3], Hsu et al. observed total mortality for inpatient and outpatient hypoglycemic T2D patients were 41 and 52/1,000 PY, respectively, which were lower than the estimates of our T2D patients with level 3 hypoglycemia. Hsu et al. recruited newly diagnosed patients with diabetes, and the diagnosis of hypoglycemia were identified by ICD diagnoses, which might have produced different results between ours.

There were discrepancies in the risk estimates evaluating the mortality of patients with mild hypoglycemia. In the studies of Mayo Clinic [2] and the Hong Kong [7], the authors noted that recurrent mild hypoglycemia was not associated with all-cause mortality despite their longer duration of diabetes and higher frequency of vascular complications. However, in the ADVANCE [11] and Hoorn Diabetes Care System Cohort, the Netherlands studies [8], there were reduced risks of all-cause and cardiovascular mortality in patients with minor hypoglycemia. In our study, the risks of all-cause and cardiovascular mortality for patients with level 1 and level 2 hypoglycemia was also inconsequential after adjustment for confounders compared with those patients with no hypoglycemia.

On the contrary, patients with self-reported severe hypoglycemia in Mayo's clinic's study [2] increased the odds of death (OR: 3.38; 95% CI 1.55–7.39) at 5 years. Studies from China [5], Taiwan [3], and Korea [4] also reported that patients with T2D who experienced inpatient, outpatient, or emergency records with an ICD diagnosis of severe hypoglycemia had a higher risk of all-cause mortality (OR: 1.76, HR: 3.49, and 3.28, respectively). Similarly, patients encountering severe hypoglycemia increased the risks of all-cause (HR: 2.69; 95% CI 1.97–3.67) and cardiovascular mortality (HR: 2.68; 95% CI 1.72–4.19) in the ADVANCE study [11]. The risk estimates of our patients with level 3 hypoglycemia documented in the emergency medical record (HR: 1.18; 95% CI 1.04–1.33) were lower than those of the above studies, but comparable to those participants who reported hypoglycemic events requiring medical assistance in the ACCORD study (HR: 1.25; 1.03–1.52) [10]. In the Hoorn Study [8], however, patients reporting severe hypoglycemic sensation was not associated with an elevated mortality risk (OR: 0.69; 95% CI 0.31–1.54). Direct comparison of the risks of all-cause and cardiovascular mortality between our results and those of previous reports might be difficult because of the different demographics, definitions of hypoglycemia and duration of study period. In contrast to the ADVANCE study [11], we did not observe any association of hypoglycemia and cardiovascular mortality. In the UK study, the authors reported that non-cardiovascular events accounted for the majority (~71%) of all deaths in both subjects with and without hypoglycemia [6].

Some authors indicated that mortality related to hypoglycemia might have been due to complexities of associated illness rather than direct causation by hypoglycemic agents [19]. In that study, hypoglycemia was classified into drug-associated or spontaneous hypoglycemia depending on hypoglycemia occurring with and without the usage of antidiabetic agents. In-hospital mortality was higher in those with spontaneous hypoglycemia (HR: 2.62; 95% CI 1.97–3.47), but the significance became null after adjustment of various confounders (HR: 1.11; 95% CI 0.76–1.64). Patients with drug-associated hypoglycemia, however, was not related to in-hospital mortality (HR: 1.06; 95% CI 0.74–1.52).

Similarly, compared with hospitalized patients with insulin-treated hypoglycemia, those with non-hypoglycemic agent related hypoglycemia had higher in-hospital mortality reported in other studies [20, 21] revealing the fact that increased mortality in hypoglycemic patients might have merely reflected the severity of the underlying illness [20]. In our study, T2D patients with level 3 hypoglycemia, i.e., severe hypoglycemia was associated with the highest risk of all-cause mortality, and that mortality persisted, albeit attenuated after adjustment for

medications, comorbidities, complications, and laboratory results. This finding suggests that, apart from severe hypoglycemia, underlying diabetes complications and concurrent chronic diseases might have contributed to the increased risk of all-cause mortality.

We observed that those patients with mean HbA1c <6.0% consisted of minor portions of all levels of hypoglycemia (~4%). HbA1c value might be underestimated in patients with diabetes in the status of advanced chronic kidney disease and anemia, and hypoglycemia was more frequent in those with mean HbA1c 7.0–9.9% rather than in those with mean HbA1c <6.0% in our previous study [12]. Similar to our findings, in the diabetes and aging study of Kaiser Permanente North California integrated healthcare delivery system, hypoglycemia was common in all levels of glycemic control, and the authors did not find patients with the lowest HbA1c levels were at high risk of hypoglycemia [22]. In the ACCORD study [23], every 1% unit higher in the updated average HbA1c concentration, there was a 15% elevated risk of hypoglycemia requiring medical assistance (HR:1.15; 95% CI 1.02–1.29) indicating patients with poorer glycemic control had a greater risk of hypoglycemia, irrespective of intensive and standard treatment groups.

To our best knowledge, studies reporting the relationship of HbA1c and mortality in patients with different levels of hypoglycemia were scarce in the literature. In a multicenter retrospective survey in China, among 590 patients with diabetes admitted for severe hypoglycemia, the mortality rates were not statistically significant between those with HbA1c <6.5% and HbA1c ≥6.5% (5.3 vs. 5.1%, P 0.17) [24]. In our study, there was no difference in the risk of all-cause mortality among patients with level 1 and level 2 hypoglycemia with any mean HbA1c. In those with level 3 hypoglycemia patients, only those who had a mean HbA1c ≥9.0% were associated with a higher risk of all-cause mortality, suggesting that high glucose variability might have been associated with increased all-cause mortality [13]. Furthermore, patients with level 1 hypoglycemia whose mean HbA1c was ≥9.0% had a significant risk of cardiovascular mortality in our T2D patients. In the ACCORD study, although the incidence of severe hypoglycemia was three times greater in the intensive than in the standard group, those in the intensive arm had a lower risk of death than those in the standard arm. The frequency of hypoglycemic episodes was also inversely related to the risk of mortality among participants in the intensive group [25]. Moreover, participants who continued to have an average HbA1c >7% in the ACCORD study seemed to be at greater risk of mortality than those with an average HbA1c <7% [26]. The findings of the above studies including our suggested that all-cause and cardiovascular mortality might have been related to poor glycemic control in addition to the consequence of hypoglycemia events.

The possible mechanisms of increased all-cause or cardiovascular mortality in hypoglycemic patients whose HbA1c ≥9.0% has not been clearly elucidated. In an Israel study [27], the authors also noted that the patients with uncontrolled glucose and documented hypoglycemia had the worst 1-year survival. There are evidences that antecedent hypoglycemia were associated with impaired autonomic function [28], cardiac rate/rhythm disturbances [29], and compromised endothelial dysfunction [30] which might all have connected to mortality in patients with T2D and hypoglycemia. However, some argued that hypoglycemia might have been a marker of susceptibility to a broad spectrum of adverse clinical conditions [11], and majority of patients with diabetes admitted for hypoglycemia died of non-cardiovascular diseases [6].

The strength of this study is the retrieval of comorbidities and complications from an electronic database might have reduced the selection and information biases. Secondly, we could assess the long-term records of various laboratory results including accurate glucose values which assisted in discerning different levels of hypoglycemia in real-life diabetes management. Thirdly, recruiting patients with anti-diabetic agents might have largely eliminated the misclassification bias in our study. However, our study is limited by the inability to detect BMI,

smoking, and blood pressure. We nevertheless have already adjusted several comorbidities, complications, medications, and laboratory results in our models. Our hypoglycemia events were based on the FEMH laboratory data, and we might have missed episodes of hypoglycemia taken placed other than in FEMH hospital settings, which might have resulted in an underestimation of hypoglycemia events. However, objective documentation of low blood glucose levels in our study could have eliminated the potential misclassification of patients with "normal feeling of hunger" and "pseudo-hypoglycemia".

## Conclusions

Our study revealed that only those patients with level 3 hypoglycemia marginally increased all-cause mortality, but not cardiovascular mortality. Hypoglycemia in patients with higher mean HbA1c rather than lower mean HbA1c were associated with an elevated risk of all-cause or cardiovascular mortality. Our study reflects the current view that obtaining an optimal HbA1c goal and reducing glucose variability might be beneficial in daily clinical practice. Aggressive treatment of underlying comorbidities and effective health education to prevent severe hypoglycemia may decrease mortality rates.

## Supporting information

**S1 Table. Type of diabetes and its comorbidities and complications collected in this study.** (DOCX)

## Author Contributions

**Conceptualization:** Seng-Wei Ooi, Hua-Fen Chen.

**Data curation:** Seng-Wei Ooi.

**Formal analysis:** Ya-Hui Chang, Hua-Fen Chen.

**Writing – original draft:** Seng-Wei Ooi, Shu-Tin Yeh, Hua-Fen Chen.

**Writing – review & editing:** Shu-Tin Yeh, Chung-Yi Li, Hua-Fen Chen.

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
