## [Decision Letter · Decision Letter 0]

22 May 2023

PONE-D-23-02045Different levels of hypoglycemia in patients with type 2 diabetes, their achieved mean HbA1c vs. all-cause and cardiovascular mortalityPLOS ONE

Dear Dr. Author,

Thank you for submitting your manuscript to PLOS ONE. After careful consideration, we feel that it has merit but does not fully meet PLOS ONE’s publication criteria as it currently stands. Therefore, we invite you to submit a revised version of the manuscript that addresses the points raised during the review process.

ACADEMIC EDITOR:

Please review the attached reviewer comments and address the concerns 

We look forward to receiving your revised manuscript.

Kind regards,

Fateen Ata, MD

Academic Editor

PLOS ONE

Journal Requirements:

- 10.7717/peerj.14609

- https://doi.org/10.1371/journal.pone.0272137

- https://doi.org/10.1016/j.amjmed.2011.07.011

In your revision ensure you cite all your sources (including your own works), and quote or rephrase any duplicated text outside the methods section. Further consideration is dependent on these concerns being addressed.

Reviewers' comments:

Reviewer's Responses to Questions

**Comments to the Author**

1. Is the manuscript technically sound, and do the data support the conclusions?

Reviewer #1: Yes

Reviewer #2: Yes

Reviewer #3: Yes

2. Has the statistical analysis been performed appropriately and rigorously? 

Reviewer #1: I Don't Know

Reviewer #2: Yes

Reviewer #3: Yes

3. Have the authors made all data underlying the findings in their manuscript fully available?

Reviewer #1: Yes

Reviewer #2: Yes

Reviewer #3: Yes

4. Is the manuscript presented in an intelligible fashion and written in standard English?

Reviewer #1: Yes

Reviewer #2: Yes

Reviewer #3: Yes

5. Review Comments to the Author

Reviewer #1: Seng-Wei Ooi et al. investigated the relationship between several levels of hypoglycemia and all-cause or cardiovascular mortality and between mean HbA1c levels and all-cause or cardiovascular mortality in 27,932 patients with T2D in Taiwan (2008-2018). They found that significant increased risk of all-cause mortality was associated with only most severe level 3 hypoglycemia levels in patients with T2D. In addition, T2D patients with mean HbA1c ≥ 9% manifested significant higher all-cause and cardiovascular mortality than T2D patients with mean HbA1c < 9%.

This study was well conducted and the observed results provided considerable clinical significance in management of patients with T2D.

Since serum lipid disorders are critical problem for the development of cardiovascular events and its mortality, in addition to LDL-C, data of triglycerides and HDL-C are needed for the assessment.

Reviewer #2: Thanks authors for carried out this study . there are some concern which authors mentioned in limitation which is acceptable

I would suggest author to bring more pathophysiological explanations about how hypoglycemic could increase the risk of mortality ; in discussion section only other's studies result compared , but needs more explanations

Reviewer #3: The paper showed the relationship between various levels of hypoglycemia and mean glycated hemoglobin A1c (HbA1c) achieved with all-cause and cardiovascular mortality in patients with T2D.They reveal that patients with level 3 hypoglycemia marginally increased all-cause mortality, but not cardiovascular mortality. Hypoglycemia in patients with higher mean HbA1c rather than low mean HbA1c were associated with an elevated risk of all-cause or cardiovascular mortality, which is of interest.

6. PLOS authors have the option to publish the peer review history of their article (what does this mean?). If published, this will include your full peer review and any attached files.

Reviewer #1: No

Reviewer #2: No

Reviewer #3: No

---

## [Author Response · Author response to Decision Letter 0]

15 Jun 2023

Response to the editor and the reviewers

Response to the editor

Authors’ reply

We tried our best to revise the manuscript according to PLOS ONE's style requirements.

1. 10.7717/peerj.14609 (shaded with blue)

2. https://doi.org/10.1371/journal.pone.0272137 (shaded with yellow)

3. https://doi.org/10.1016/j.amjmed.2011.07.011 (shaded with green)

In your revision ensure you cite all your sources (including your own works), and quote or rephrase any duplicated text outside the methods section. Further consideration is dependent on these concerns being addressed.

Authors’ reply

We used Microsoft Word to compare the current manuscript to our previous works with another manuscript, and to our best rephrase the duplicate text. We quoted all the works above mentioned (reference 14, 12, and 19 of the revised manuscript)

Authors’ reply

We deleted the ethics statement apart from the Methods section (Please see the page 23, Lines 471-473 of the revised manuscript), and amended the ethics statement including the full name of the ethics committee/institutional review board(s) in methods section (Please see the page 5, Lines 116-118 of the revised manuscript). 

Authors’ reply

The data sets analyzed during the current study are not publicly available because the restrictions are imposed by the Research Ethics Review Committee of Far Eastern Memorial Hospital. Although we cannot share the data publicly, the minimal datasets are available upon request from the Research Ethics Review Committee of Far Eastern Memorial Hospital repository (via: https://www.femh-irb.org/index.php/procedure). Application to the Research Ethics Review Committee of Far Eastern Memorial Hospital (contact via irb@mail.femh.org.tw, reference number 108008-F) is available for researchers.

Response to the reviewer 1

1. Seng-Wei Ooi et al. investigated the relationship between several levels of hypoglycemia and all-cause or cardiovascular mortality and between mean HbA1c levels and all-cause or cardiovascular mortality in 27,932 patients with T2D in Taiwan (2008-2018). They found that significant increased risk of all-cause mortality was associated with only most severe level 3 hypoglycemia levels in patients with T2D. In addition, T2D patients with mean HbA1c ≥ 9% manifested significant higher all-cause and cardiovascular mortality than T2D patients with mean HbA1c < 9%. 

This study was well conducted and the observed results provided considerable clinical significance in management of patients with T2D. 

Since serum lipid disorders are critical problem for the development of cardiovascular events and its mortality, in addition to LDL-C, data of triglycerides and HDL-C are needed for the assessment.

Authors’ reply

We added data of triglycerides and HDL-C (Please see Page 6, Lines 147-148, Page 11, Table 1, of the revised manuscript) We also re-analyzed all the regression models with addition of triglycerides and HDL-C, and the results are presented in Tables 2 and 3 (Please see Page 13-14, Table 2 and Page 16-17, Table 3 of the revised manuscript). 

The response to the reviewer 2

1. Thanks authors for carried out this study . there are some concern which authors mentioned in limitation which is acceptable. 

I would suggest author to bring more pathophysiological explanations about how hypoglycemic could increase the risk of mortality ; in discussion section only other's studies result compared , but needs more explanations

Authors’ reply

We have revised the discussion section with addition of plausible pathophysiological mechanisms about how hypoglycemic could increase the risk of mortality. (Page 21, Lines 429-438 of the revised manuscript)

The response to the reviewer 3

1. The paper showed the relationship between various levels of hypoglycemia and mean glycated hemoglobin A1c (HbA1c) achieved with all-cause and cardiovascular mortality in patients with T2D.They reveal that patients with level 3 hypoglycemia marginally increased all-cause mortality, but not cardiovascular mortality. Hypoglycemia in patients with higher mean HbA1c rather than low mean HbA1c were associated with an elevated risk of all-cause or cardiovascular mortality, which is of interest.

---

## [Decision Letter · Decision Letter 1]

26 Jun 2023

Different levels of hypoglycemia in patients with type 2 diabetes, their achieved mean HbA1c vs. all-cause and cardiovascular mortality

PONE-D-23-02045R1

Dear Dr. Chen

We’re pleased to inform you that your manuscript has been judged scientifically suitable for publication and will be formally accepted for publication once it meets all outstanding technical requirements.

Kind regards,

Fateen Ata, MD

Academic Editor

PLOS ONE

Additional Editor Comments (optional):

All revisions by the reviewers have been considered and made by the authors in a satisfactory manner. 

Reviewers' comments:

Reviewer's Responses to Questions

**Comments to the Author**

1. If the authors have adequately addressed your comments raised in a previous round of review and you feel that this manuscript is now acceptable for publication, you may indicate that here to bypass the “Comments to the Author” section, enter your conflict of interest statement in the “Confidential to Editor” section, and submit your "Accept" recommendation.

Reviewer #1: All comments have been addressed

2. Is the manuscript technically sound, and do the data support the conclusions?

Reviewer #1: Yes

3. Has the statistical analysis been performed appropriately and rigorously? 

Reviewer #1: Yes

4. Have the authors made all data underlying the findings in their manuscript fully available?

Reviewer #1: No

5. Is the manuscript presented in an intelligible fashion and written in standard English?

Reviewer #1: Yes

6. Review Comments to the Author

Reviewer #1: (No Response)

7. PLOS authors have the option to publish the peer review history of their article (what does this mean?). If published, this will include your full peer review and any attached files.

Reviewer #1: **Yes: **Ken-ichi Aihara

---

## [Editor Report · Acceptance letter]

5 Jul 2023

PONE-D-23-02045R1 

Different levels of hypoglycemia in patients with type 2 diabetes, their achieved mean HbA1c vs. all-cause and cardiovascular mortality 

Dear Dr. Chen:

I'm pleased to inform you that your manuscript has been deemed suitable for publication in PLOS ONE. Congratulations! Your manuscript is now with our production department. 

Kind regards, 

on behalf of

Dr. Fateen Ata 

Academic Editor

PLOS ONE